# REMEMBERING FOR THE RIGHT REASONS: EXPLANATIONS REDUCE CATASTROPHIC FORGETTING

**Sayna Ebrahimi**[1], **Suzanne Petryk**[1], **Akash Gokul**[1], **William Gan**[1], **Joseph E. Gonzalez**[1], **Marcus Rohrbach**[2], **Trevor Darrell**[1]

[1]UC Berkeley, [2] Facebook AI Research

{sayna,spetryk,akashgokul,wjgan,jegonzal,trevordarrell}@berkeley.edu
mrf@fb.com

## ABSTRACT

The goal of continual learning (CL) is to learn a sequence of tasks without suffering from the phenomenon of catastrophic forgetting. Previous work has shown that leveraging memory in the form of a replay buffer can reduce performance degradation on prior tasks. We hypothesize that forgetting can be further reduced when the model is encouraged to remember the *evidence* for previously made decisions. As a first step towards exploring this hypothesis, we propose a simple novel training paradigm, called Remembering for the Right Reasons (RRR), that additionally stores visual model explanations for each example in the buffer and ensures the model has "the right reasons" for its predictions by encouraging its explanations to remain consistent with those used to make decisions at training time. Without this constraint, there is a drift in explanations and increase in forgetting as conventional continual learning algorithms learn new tasks. We demonstrate how RRR can be easily added to any memory or regularization-based approach and results in reduced forgetting, and more importantly, improved model explanations. We have evaluated our approach in the standard and few-shot settings and observed a consistent improvement across various CL approaches using different architectures and techniques to generate model explanations and demonstrated our approach showing a promising connection between explainability and continual learning. Our code is available at https://github.com/SaynaEbrahimi/Remembering-for-the-Right-Reasons.

## 1 INTRODUCTION

Humans are capable of continuously learning novel tasks by leveraging their lifetime knowledge and expanding them when they encounter a new experience. They can remember the majority of their prior knowledge despite the never-ending nature of their learning process by simply keeping a running tally of the observations distributed over time or presented in summary form. The field of continual learning or lifelong learning (Thrun & Mitchell, 1995; Silver et al., 2013) aims at maintaining previous performance and avoiding so-called catastrophic forgetting of previous experience (McCloskey & Cohen, 1989; McClelland et al., 1995) when learning new skills. The goal is to develop algorithms that continually update or add parameters to accommodate an online stream of data over time.

An active line of research in continual learning explores the effectiveness of using small memory budgets to store data points from the training set (Castro et al., 2018; Rajasegaran et al., 2020; Rebuffi et al., 2017; Wu et al., 2019), gradients (Lopez-Paz et al., 2017), or storing an online generative model that can fake them later (Shin et al., 2017). Memory has been also exploited in the form of accommodating space for architecture growth and storage to fully recover the old performance when needed (Ebrahimi et al., 2020b; Rusu et al., 2016). Some methods store an old snapshot of the model to distill the features (Li & Hoiem, 2016) or attention maps (Dhar et al., 2019) between the teacher and student models.

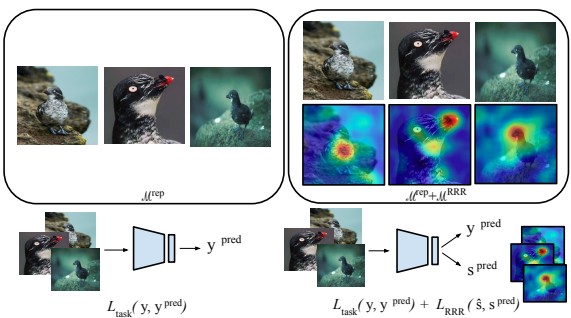

Figure 1: An illustration of applying RRR paradigm. **(Left)** In a typical experience replay scenario, samples from prior tasks are kept in a memory buffer $\mathcal{M}^{\text{rep}}$ and revisited during training. **(Right)** In our proposed idea (RRR), in addition to $\mathcal{M}^{\text{rep}}$, we also store model explanations (saliency maps) as $\mathcal{M}^{\text{RRR}}$ for those samples and encourage the model to remember the original reasoning for the prediction. Note that the saliency maps are small masks resulting in a negligible memory overhead (see Section 4.1).

The internal reasoning process of deep models is often treated as a black box and remains hidden from the user. However, recent work in explainable artificial intelligence (XAI) has developed methods to create human-interpretable explanations for model decisions (Simonyan et al., 2013; Zhang et al., 2018; Petsiuk et al., 2018; Zhou et al., 2016; Selvaraju et al., 2017). We posit that the catastrophic forgetting phenomenon is due in part to not being able to rely on the same reasoning as was used for a previously seen observation. Therefore, we hypothesize that forgetting can be mitigated when the model is encouraged to remember the *evidence* for previously made decisions. In other words, a model which can remember its final decision and can reconstruct the same prior reasoning. Based on this approach, we develop a novel strategy to exploit explainable models for improving performance.

Among the various explainability techniques proposed in XAI, saliency methods have emerged as a popular tool to identify the support of a model prediction in terms of relevant features in the input. These methods produce saliency maps, defined as regions of visual evidence upon which a network makes a decision. Our goal is to investigate whether augmenting experience replay with explanation replay reduces forgetting and how enforcing to remember the explanations will affect the explanations themselves. Figure 1 illustrates our proposed method.

In this work, we propose RRR, a training strategy guided by model explanations generated by any *white-box* differentiable explanation method; RRR adds an *explanation loss* to continual learning. White-box methods generate an explanation by using some internal state of the model, such as gradients, enabling their use in end-to-end training. We evaluate our approach using various popular explanation methods including vanilla backpropagation (Zeiler & Fergus, 2014), backpropagation with smoothing gradients (Smoothgrad) (Smilkov et al., 2017), Guided Backpropagation (Springenberg et al., 2014), and Gradient Class Activation Mapping (Grad-CAM) (Selvaraju et al., 2017) and compare their performance versus their computational feasibility. We integrate RRR into several state of the art class incremental learning (CIL) methods, including iTAML (Rajasegaran et al., 2020), EEIl (Castro et al., 2018), BiC (Wu et al., 2019), TOPIC (Tao et al., 2020), iCaRL (Rebuffi et al., 2017), EWC (Kirkpatrick et al., 2017), and LwF (Li & Hoiem, 2016). Note that RRR does not require task IDs at test time. We qualitatively and quantitatively analyze model explanations in the form of saliency maps and demonstrate that RRR remembers its earlier decisions in a sequence of tasks due to the requirement to focus on the the right evidence. We empirically show the effect of RRR in standard and few-shot class incremental learning (CIL) scenarios on popular benchmark datasets including CIFAR100, ImageNet100, and Caltech-UCSD Birds 200 using different network architectures where RRR improves overall accuracy and forgetting over experience replay and other memory-based method.

Our contribution is threefold: we first propose our novel, simple, yet effective memory constraint, which we call Remembering for the Right Reasons (RRR), and show that it reduces catastrophic forgetting by encouraging the model to look at the same explanations it initially found for its decisions. Second, we show how RRR can be readily combined with memory-based and regularization-based

CL methods to improve performance. Third, we demonstrate how guiding a continual learner to remember its explanations can improve the quality of the explanations themselves; i.e., the model looks at the right region in an image when making correct decisions while it focuses its maximum attention on the background when it misclassifies an object.

## 2 BACKGROUND: WHITE-BOX EXPLANABILITY TECHNIQUES

Here we briefly review the explainability methods we have evaluated our approach with. The core idea behind RRR is to guide explanations or saliency maps during training to preserve their values. Hence, only gradient-based saliency techniques can be used which are differentiable and hence trainable *during* training for the mainstream task as opposed to black-box saliency methods which can be used only *after* training to determine important parts of an image.

**Vanilla Backpropagation** (Zeiler & Fergus, 2014): The simplest way to understand and visualize which pixels are most salient in an image is to look at the gradients. This is typically done by making a forward pass through the model and taking the gradient of the given output class with respect to the input. Those pixel-wise derivative values can be rendered as a normalized heatmap representing the amount of change in the output probability of a class caused by perturbing that pixel. To store a saliency map for each RGB image of size $3 \times W \times H$, we need an equivalent memory size of storing $W \times H$ pixel values.

**Backpropagation with SmoothGrad**: Smilkov et al. (2017) showed that the saliency maps obtained using raw gradients are visually noisy and using them as a proxy for feature importance is sub-optimal. They proposed a simple technique for denoising the gradients that adds pixel-wise Gaussian noise to $n$ copies of the image, and simply averages the resulting gradients. SmoothGrad requires the same amount of memory to store the saliency maps while it takes $n$ times longer to repeat generating each saliency map. We found $n = 40$ to be large enough to make a noticeable change in the saliencies in our experiments.

**Gradient-weighted Class Activation Mapping (Grad-CAM)** (Selvaraju et al., 2017): is a white-box explainability technique which uses gradients to determine the influence of specific feature map activations on a given prediction. Because later layers in a convolutional neural network are known to encode higher-level semantics, taking the gradient of a model output with respect to the activations of these feature maps discovers which high-level semantics are important for the prediction. We refer to this layer as the *target layer* in our analysis. For example, when using Grad-CAM to visualize explanations for image classification, taking the gradient of the correct class prediction with respect to the last convolutional layer highlights class-discriminative regions in the image (such as the wings of a bird when identifying bird species).

Consider the pre-softmax score $y_c$ for class $c$ in an image classification output. In general, any differentiable activation can be used. Consider also a single convolutional layer with $K$ feature maps, with a single feature map noted as $A^k \in \mathbb{R}^{u \times v}$. Grad-CAM takes the derivative of $y_c$ with respect to each feature map $A^k$. It then performs global average pooling over the height and width dimensions for each of these feature map gradients, getting a vector of length $K$. Each element in this vector is used as a weight $\alpha_k^c$, indicating the importance of feature map $k$ for the prediction $y_c$. Next, these importance weights are used in computing a linear combination of the feature maps. Followed by a ReLU (Jarrett et al., 2009) to zero-out any activations with a negative influence on the prediction of class $c$, the final Grad-CAM output ($s$) is as below with $A_{ij}^k$ defined at location $(i, j)$ in feature map $A^k$.

$$\alpha_k^c = \quad \frac{1}{uv} \sum_{i=1}^{u} \sum_{j=1}^{v} \frac{\partial y_c}{\partial A_{ij}^k} \qquad s_{Grad\text{-}CAM}^c = ReLU \left( \sum_{k=1}^{K} \alpha_k^c A^k \right) \tag{1}$$

Unlike the common saliency map techniques of Guided BackProp (Springenberg et al., 2014), Guided GradCAM (Selvaraju et al., 2016), Integrated Gradients (Sundararajan et al., 2017b), Gradient $\odot$ Input (Shrikumar et al., 2016), Backpropagation with SmoothGrad (Smilkov et al., 2017) etc., vanilla Backpropagation and Grad-CAM pass important "sanity checks" regarding their sensitivity to data and model parameters (Adebayo et al., 2018). We will compare using vanilla Backpropagation, Backpropagation with SmoothGrad, and Grad-CAM in RRR in Section 4. We will refer to the function that computes the output $s$ of these saliency method as $\mathcal{XAI}$.

---

**Algorithm 1** Remembering for the Right Reasons (RRR) for Continual Learning

---

1: **function** TRAIN ($f_\theta, \mathcal{D}^{tr}, \mathcal{D}^{ts}$)
2:     $T$: # of tasks, $n$: # of samples in task
3:     $R \leftarrow 0 \in \mathbb{R}^{T \times T}$
4:     $\mathcal{M}^{\text{rep}} \leftarrow \{\}$
5:     $\mathcal{M}^{\text{RRR}} \leftarrow \{\}$
6:     **for** $k = 1$ to T **do**
7:         **for** $i = 1$ to n **do**
8:             Compute cross entropy on task ($\mathcal{L}_{\text{task}}$)
9:             Compute $\mathcal{L}_{\text{RRR}}$ using Eq. 2
10:             $\theta' \leftarrow \theta - \alpha \nabla_\theta (\mathcal{L}_{\text{task}} + \mathcal{L}_{\text{RRR}})$
11:         **end for**
12:         $\mathcal{M}^{\text{rep}}, \mathcal{M}^{\text{RRR}} \leftarrow$ UPDATE MEM($f_\theta^k, \mathcal{D}_k^{tr}, \mathcal{M}^{\text{rep}},$
                            $\mathcal{M}^{\text{RRR}}$)
13:         $R_{k,\{1\cdots k\}} \leftarrow$ EVAL ($f_\theta^k, \mathcal{D}_{\{1\cdots k\}}^{ts}$)
14:     **end for**
15:     **return** $f_\theta, R$
16: **end function**

**function** UPDATE MEM($f_\theta^k, \mathcal{D}_k^{tr}, \mathcal{M}^{\text{rep}}, \mathcal{M}^{\text{RRR}}$)
    $(x_i, k, y_i) \sim \mathcal{D}_k^{tr}$
    $\mathcal{M}^{\text{rep}} \leftarrow \mathcal{M}^{\text{rep}} \cup \{(x_i, k, y_i)\}$
    $\hat{s} \leftarrow \mathcal{XAI}(f_\theta^k(x_i, k))$
    $\mathcal{M}^{\text{RRR}} \leftarrow \mathcal{M}^{\text{RRR}} \cup \{\hat{s}\}$
    **return** $\mathcal{M}^{\text{rep}}, \mathcal{M}^{\text{RRR}}$
**end function**

**function** EVAL($f_\theta^k, \mathcal{D}_{\{1\cdots k\}}^{ts}$)
    **for** $i = 1$ to $k$ **do**
        $R_{k,i} = \text{Accuracy}(f_\theta^k(x,i), y | \forall (x,y) \in \mathcal{D}_i^{ts})$
    **end for**
    **return** $R$
**end function**

---

## 3   REMEMBERING FOR THE RIGHT REASONS (RRR)

Memory-based methods in continual learning have achieved high performance on vision benchmarks using a small amount of memory, i.e. storing a few samples from the training data into the *replay buffer* to directly train with them when learning new tasks. This simple method, known as experience replay, has been explored and shown to be effective (Rebuffi et al., 2017; Wu et al., 2019; Castro et al., 2018; Rajasegaran et al., 2020; Ebrahimi et al., 2020b; Hayes et al., 2019; Riemer et al., 2018). In this work we aim to go one step further and investigate the role of *explanations* in continual learning, particularly on mitigating forgetting and change of model explanations.

We consider the problem of learning a sequence of $T$ data distributions $\mathcal{D}^{tr} = \{\mathcal{D}_1^{tr}, \cdots, \mathcal{D}_T^{tr}\}$, where $\mathcal{D}_k^{tr} = \{(x_i^k, y_i^k)_{i=1}^{n_k}\}$ is the data distribution for task $k$ with $n$ sample tuples of input ($\mathbf{x}^k \subset \mathcal{X}$) and set of output labels ($\mathbf{y}^k \subset \mathcal{Y}$). The goal is to sequentially learn the model $f_\theta : \mathcal{X} \times \mathcal{T} \rightarrow \mathcal{Y}$ for each task that can map each input $x$ to its target output, $y$, while maintaining its performance on all prior tasks. We aim to achieve this by using memory to enhance better knowledge transfer as well as better avoidance of catastrophic forgetting. We assume two limited memory pools $\mathcal{M}^{\text{rep}}$ for raw samples and $\mathcal{M}^{\text{RRR}}$ for model explanations. In particular, $\mathcal{M}^{\text{rep}} = \{(x_i^j, y_i^j)_{i=1}^m \sim \mathcal{D}_{j=1\cdots k-1}^{tr}\}$ stores $m$ samples in total from all prior tasks to $k$. Similarly $\mathcal{M}^{\text{RRR}}$ stores the saliency maps generated based on $f_\theta^k$ by one of the explanation methods ($\mathcal{XAI}$) discussed in Section 2 for images in $\mathcal{M}^{\text{rep}}$ where $f_\theta^k$ is $f_\theta$ being trained for task $k$. We use a single-head architecture where the task ID integer $t$ is not required at test time.

Upon finishing the $k^{th}$ task, we randomly select $m/(k-1)$ samples per task from its training data and update our replay buffer memory $\mathcal{M}^{\text{rep}}$. RRR uses model explanations on memory samples to perform continual learning such that the model preserves its reasoning for previously seen observations. We explore several explanation techniques to compute saliency maps using $f_\theta^k$ for the stored samples in the replay buffer to populate the xai buffer memory $\mathcal{M}^{\text{xai}}$. The stored saliency maps will serve as reference explanations during the learning of future tasks to prevent model parameters from being altered resulting in different reasoning for the same samples. We implement RRR using an L1 loss on the error in saliency maps generated after training a new task with respect to their stored *reference* evidence.

$$\mathcal{L}_{\text{RRR}}(f_\theta, \mathcal{M}^{\text{rep}}, \mathcal{M}^{\text{RRR}}) = \mathbb{E}_{((x,y),\hat{s}) \sim (\mathcal{M}^{\text{rep}}, \mathcal{M}^{\text{RRR}})} ||\mathcal{XAI}(f_\theta^k(x)) - \hat{s}||_1 \qquad (2)$$

where $\mathcal{XAI}(\cdot)$ denotes the explanation method used to compute saliency maps using the model trained on the last seen example from task $k$, and $\hat{s}$ are the *reference* saliency maps generated by $\mathcal{XAI}(f_\theta^k)$ upon learning each task prior to $T$ and stored in to the memory. We show below that combining RRR into the objective function of state-of-the-art memory and regularization-based methods results in significant performance improvements. The full algorithm for RRR is given in Alg. 1.

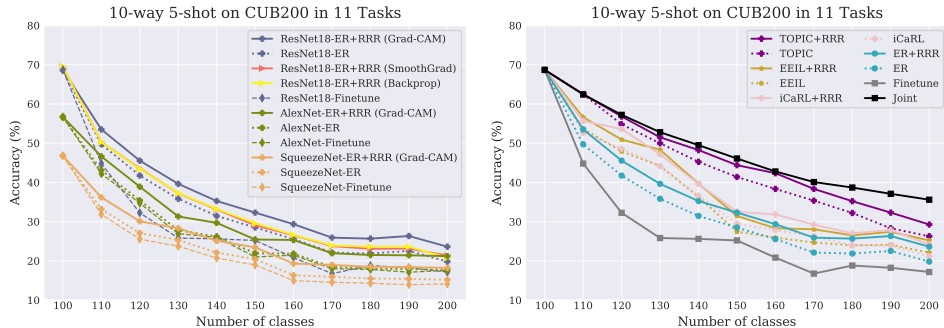

Figure 2: Few-shot CIL learning of CUB200 in 11 tasks where each point shows the classification accuracy on all seen classes so far. (**Left**) Shows ER with and without $\mathcal{L}_{\text{RRR}}$ using different back-bone architectures and saliency map techniques. (**Right**) Performance of the state-of-the-art existing approaches with and without $\mathcal{L}_{\text{RRR}}$ on CUB200 including TOPIC (Tao et al., 2020), EEIL (Castro et al., 2018), iCaRL (Rebuffi et al., 2017). Joint training serves as the upper bound. Results for baselines are obtained using their original implementation. All results are averaged over 3 runs and mean and standard deviation values are given in the appendix. Best viewed in color.

## 4 EXPERIMENTS

In this section, we apply RRR in two distinct learning regimes: standard and few-shot class incremental learning. These are the most challenging CL scenarios, in which task descriptions are not available at test time. We first explore the effect of backbone architecture and the saliency map technique on RRR performance. We then report our obtained results integrating $\mathcal{L}_{\text{RRR}}$ into existing memory-based and regularization-based methods.

### 4.1 FEW-SHOT CIL PERFORMANCE

We first explore CIL of low-data regimes where preventing overfitting to few-shot new classes is another challenge to overcome in addition to avoiding catastrophic forgetting of old classes. We use $C$ classes and $K$ training samples per class as the $C$-way $K$-shot few-shot class incrementally learning setting where we have a set of $b$ base classes to learn as the first task while the remaining classes are learned with only a few randomly selected samples. In order to provide a direct comparison to the state-of-the-art work of Tao et al. (2020) we precisely followed their setup and and used the same Caltech-UCSD Birds dataset (Wah et al., 2011), divided into 11 disjoint tasks and a 10-way 5-shot setting, where the first task contains $b = 100$ base classes resulting in 3000 samples for training and 2834 images for testing. The remaining 100 classes are divided into 10 tasks where 5 samples per class are randomly selected as the training set, while the test set is kept intact containing near 300 images per task. The images in CUB200 are resized to $256 \times 256$ and then randomly cropped to $224 \times 224$ for training. We store 4 images per class from base classes in the first task and 1 sample per each few-shot class in the remaining 10 tasks (Tao et al., 2020). We used the RAdam (Liu et al., 2019) optimizer with a learning rate of 0.001 which was reduced by 0.2 at epochs 20, 40, and 60 and trained for a total of 70 epochs with a batch size of 128 for the first and 10 for the remaining tasks.

Figure 2 (left) shows results for ER with and without $\mathcal{L}_{\text{RRR}}$ using different backbone architectures and saliency map techniques. Among the tested saliency map methods, Grad-CAM on ResNet18 outperforms Vanilla Backpropagation and SmoothGrad by 2-3% while SmoothGrad and vanilla Backpropagation achieve similar CL performance. To compute the memory overhead of storing the output for a saliency method, if we assume the memory required to store an image is $M$, vanilla Backpropagation and SmoothGrad generate a pixel-wise saliency map that occupies $M/3$ of memory. However, in Grad-CAM the saliency map size is equal to the feature map of the *target layer* in the architecture. In our study with Grad-CAM we chose our *target layer* to be the last convolution layer before the fully-connected layers. For instance using ResNet18 for colored $224 \times 224$ images results in the Grad-CAM output of $7 \times 7$ occupying 196B. Table 2 shows the target layer name and

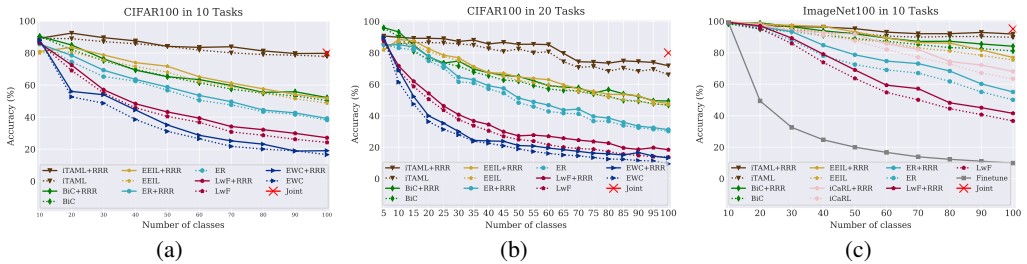

Figure 3: Effect of RRR on existing methods for CIL on CIFAR100 in (a) 10 and (b) 20 tasks and (c) ImageNet100 in 10 tasks. Each point shows the classification accuracy on all seen classes so far. Results for iTAML, BiC, and EEIL are produced with their original implementation while EWC and LwF are re-implemented by us. All results are averaged over 3 runs and mean and standard deviation values are given in the appendix. Best viewed in color.

saliency map size for other network architectures used in this work (AlexNet and SqueezeNet1_1) as well.

Figure 2 (right) shows the effect of adding $\mathcal{L}_{\text{RRR}}$ on existing recent state-of-the-art methods such as TOPIC (Tao et al., 2020), EEIL (Castro et al., 2018), and iCaRL (Rebuffi et al., 2017). Tao et al. (2020) used a neural gas network (Martinetz et al., 1991; Fritzke et al., 1995) which can learn and preserve the topology of the feature manifold formed by different classes and we have followed their experimental protocol for our CUB200 experiment by using identical samples drawn in each task which are used across all the baselines for fair comparison. Adding $\mathcal{L}_{\text{RRR}}$ improves the performance of all the baselines; TOPIC becomes nearly on-par with joint training which serves as the upper bound and does not adhere to continual learning. The gap between ER and iCaRL is also reduced when ER uses $\mathcal{L}_{\text{RRR}}$.

## 4.2 STANDARD CIL PERFORMANCE

In order to provide a direct comparison to the recent work of Rajasegaran et al. (2020) we perform our standard CIL experiment on CIFAR100 (Krizhevsky & Hinton, 2009) and ImageNet100 where we assume a memory budget of 2000 samples which are identical across all the baselines. Following Rajasegaran et al. (2020) we divide CIFAR100 to 10 and 20 disjoint tasks with 10 and 5 classes at a time. Figures 3a and 3b show the classification accuracy upon learning each task on all seen classes. We see a consistent average improvement of $2 - 4\%$ when $\mathcal{L}_{\text{RRR}}$ is added as an additional constraint to preserve the model explanations across all methods, from the most naive memory-based method, experience replay (ER), to more sophisticated approaches which store a set of old class exemplars along with meta-learning (iTAML), correct bias for new classes (BiC), and fine tune on the exemplar set (EEIL). We also applied the RRR constraint on regularization-based methods such as EWC and LwF with no memory used as a replay buffer. The accuracy for both improves despite not benefiting from revisiting the raw data. However, they fall behind all memory-based methods with or without $\mathcal{L}_{\text{RRR}}$. The final accuracy on the entire sequence for joint training (multi-task learning) with RAdam optimizer (Liu et al., 2019) is $80.03\%$ which serves as an upper bound as it has access to data from all tasks at all time.

Figure 3c shows our results on learning ImageNet100 in 10 tasks. The effect of adding $\mathcal{L}_{\text{RRR}}$ to baselines on the ImageNet100 experiment is more significant $(3-6\%)$ compared to CIFAR100. This is mainly due to the larger size and better quality of images in ImageNet100, resulting in generating larger Grad-CAM saliency maps. These experiments clearly reveal the effectiveness of $\mathcal{L}_{\text{RRR}}$ on model explanations in a continual learning problem at nearly zero cost of memory overhead when a memory buffer is already created and applied as a catastrophic forgetting avoidance strategy. This makes Grad-CAM the ideal approach to generate saliency maps when applying the RRR training strategy, as it achieves the highest accuracy while utilizing the least storage space to store saliencies. Note that we adopt Grad-CAM to generate saliency maps in the remaining experiments in this paper. We also keep using only ResNet18 for a fair comparison with the literature.

Table 1: PG experiment results on few-shot CIL CUB200 measuring (a) PG-ACC (%) and PG-BWT (%) and (b) precision and recall averaged over all tasks. $Pr_{i,i}$ and $Re_{i,i}$ evaluate the pointing game on each task $\mathbf{t^i}$ directly after the model has been trained on $\mathbf{t^i}$. $Pr_{T,i}$ and $Re_{T,i}$ are obtained by the evaluation for task $\mathbf{t^i}$ using the model trained for all $T$ tasks.

(a) PG localization accuracy and backward transfer

| Methods | PG-ACC (%) | PG-BWT (%) |
|---|---|---|
| ER | 54.0 | -17.4 |
| ER+RRR | 58.5 | -15.6 |
| TOPIC | 72.7 | -0.9 |
| TOPIC+RRR | 74.2 | -2.1 |

(b) Precision and recall using PG experiment

| | Precision | | Recall | |
|---|---|---|---|---|
| Methods | $Pr_{i,i}$ | $Pr_{T,i}$ | $Re_{i,i}$ | $Re_{T,i}$ |
| ER | 80.0 | 68.9 | 64.1 | 65.1 |
| ER+RRR | 82.1 | 70.3 | 64.2 | 66.8 |
| TOPIC | 91.0 | 88.4 | 98.1 | 97.4 |
| TOPIC+RRR | 92.8 | 89.1 | 99.6 | 99.2 |

## 5 ANALYSIS OF MODEL EXPLANATIONS

In this section we want to answer the question "*How often does the model remember its decision for the right reason upon learning a sequence of tasks?*". In particular, we want to evaluate how often the model is "pointing" at the right evidence for its predictions, instead of focusing its maximum attention on the background or other objects in the image. We use the Pointing Game experiment (PG) (Zhang et al., 2018) for this evaluation, which was introduced to measure the discriminativeness of a visualization method for target object localization. Here, we use ground truth segmentation annotation labels provided with the CUB-200 dataset to define the true object region.

First, we look into *hits* and *misses* defined by the PG experiment. When the location of the maximum in a predicted saliency map falls inside the object, it is considered as a *hit* and otherwise it is a *miss*. Figure 5 shows an example from CUB200 where the segmentation annotation is used to determine whether the peak of the predicted saliency map (marked with red cross) falls inside the object region. This example is regarded as *hit* as the red cross is inside the segmentation mask for the bird. PG localization accuracy is defined as the number of hits over the total number of predictions. We would like to measure both the overall PG performance of a continual learner as well as how much learning new tasks causes it to forget its ability to *hit* the target object. For these metrics, inspired by (Lopez-Paz et al., 2017), we define PG-ACC $= \frac{1}{T} \sum_{i=1}^{T} R_{T,i}$ as the average PG localization accuracy computed over all prior tasks after training for each new task and PG-BWT $= \frac{1}{T-1} \sum_{i=1}^{T-1} R_{T,i} - R_{i,i}$ (backward transfer) which indicates how much learning new tasks has influenced the PG localization accuracy on previous tasks where $R_{n,i}$ is the on task $i$ after learning the $n^{\text{th}}$ task. Results for ER and TOPIC with and without $\mathcal{L}_{\text{RRR}}$ on CUB200 are shown in Table 1a. It shows how constraining different models to remember their initial evidence can lead to better localization of the bird across learning new tasks.

However, PG performance does not capture all of our desired properties for a continual learner. Ideally, we not only want a model to predict the object correctly if it is looking at the right evidence, but also we want it to not predict an object if it is not able to find the right evidence for it. To measure how close our baselines can get to this ideal model when they are combined with $\mathcal{L}_{\text{RRR}}$, we measure the precision as $\text{tp}/(\text{tp+fp})$, and recall as $\text{tp}/(\text{tp+fn})$. We evaluate these metrics once immediately after learning each task, denoted as $Pr_{i,i}$ and $Re_{i,i}$, respectively, and again at the end of the learning process of final task $T$ denoted as $Pr_{T,i}$ and $Re_{T,i}$, where the first subscript refers to the model ID and the second subscript is the test dataset ID on which the model is evaluated. The higher the precision for a model is, the less often it has made the right decision without looking at the right evidence. On the other hand, the higher the recall, the less often it makes a wrong decision despite looking at the correct evidence. We show our evaluation on these metrics in Table 1b for ER and TOPIC with and without $\mathcal{L}_{\text{RRR}}$ on CUB200 where $\mathcal{L}_{\text{RRR}}$ increases both precision and recall across all methods, demonstrating that our approach continually makes better predictions because it finds the right evidence for its decisions.

In our final analysis, we would like to visualize the evolution of saliency maps across learning a sequence of tasks. Figure 4 illustrates the evolution of saliency maps for an image from the test-set of the second task, which both ER without $\mathcal{L}_{\text{RRR}}$ (top row) and with $\mathcal{L}_{\text{RRR}}$ (bottom row) have seen during training for the second task. We only visualize the generated saliencies after finishing tasks #2, #5, #7, #9, and #11 for simplicity. We indicate the correctness of the prediction made by each model with 'correct' or 'incorrect' written on top of their corresponding saliency map. Our goal is

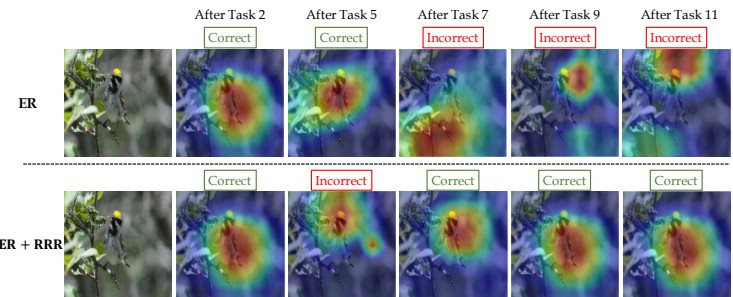

Figure 4: An illustration of the progression of saliencies on an image from the test set of the second task, evaluated after the model is trained on tasks #2, #5, #7, #9, and #11 on CUB200. Failure case for ER w.o. $\mathcal{L}_{RRR}$ (top row), where saliency drifts from the original and the prediction becomes incorrect. ER+RRR (bottom row) retains close to the original saliency as the model trains on more tasks, with the exception of Task #5 which it is able to correct later on. Its performance is retained as well, for saliencies that are close to the original.

to visualize if adding the loss term $\mathcal{L}_{RRR}$ prevents the *drifting* of explanations. Given the same input image, the ER without $\mathcal{L}_{RRR}$ model makes an incorrect prediction after being continually trained for 11 tasks while never recovering from its mistake. On the other hand, when it is combined with $\mathcal{L}_{RRR}$. it is able to recover from an early mistake after task 5. Considering the saliency map obtained after finishing task one as a *reference* evidence, we can see that ER's evidence drifts further from the reference. On the bottom row, the region of focus of ER+RRR remains nearly identical to its initial evidence, apart from one incorrect prediction. As applying $\mathcal{L}_{RRR}$ corrects its saliency back to the original, this prediction is corrected as well. This supports the conclusion that retaining the original saliency is important for retaining the original accuracy.

## 6    RELATED WORK

**Continual learning:** Past work in CL has generally made use of either memory, model structure, or regularization to prevent catastrophic forgetting. *Memory-based methods* store some form of past experience into a replay buffer. However, the definition of "experience" varies between methods. Rehearsal-based methods use episodic memories as raw samples (Robins, 1995; Rebuffi et al., 2017; Riemer et al., 2018) or their gradients (Lopez-Paz et al., 2017; Chaudhry et al., 2019) for the model to revisit. Incremental Classifier and Representation Learning (iCaRL) (Rebuffi et al., 2017), is a class-incremental learner that uses a nearest-exemplar algorithm for classification and prevents catastrophic forgetting by using an episodic memory. iTAML (Rajasegaran et al., 2020) is a task-agnostic meta-learning algorithm that uses a momentum based strategy for meta-update and in addition to the object classification task, it predicts task labels during inference. An end-to-end incremental learning framework (EEIL) (Castro et al., 2018) also uses an exemplar set along with data augmentation and balanced fine-tuning to alleviate the imbalance between the old and new classes. Bias Correction Method (BiC) (Wu et al., 2019) is another class-incremental learning algorithm for large datasets in which a linear model is used to correct bias towards new classes using a fully connected layer. In contrast, pseudo-rehearsal methods generate the replay samples using a generative model such as an autoencoder (Kemker & Kanan, 2017) or a GAN (Kamra et al., 2017; Shin et al., 2017). *Regularization-based methods* define different metrics to measure importance and limit the changes on parameters accordingly (Kirkpatrick et al., 2017; Zenke et al., 2017; Ebrahimi et al., 2020a; Serra et al., 2018; Li & Hoiem, 2016; Dhar et al., 2019) but in general these methods have limited capacity. *Structure-based methods* control which portions of a model are responsible for specific tasks such that the model increases its capacity in a controlled fashion as more tasks are added. Inference for different tasks can be restricted to various neurons (Fernando et al., 2017; Yoon et al., 2018), columns (Rusu et al., 2016), task-specific modules (Ebrahimi et al., 2020b), or parameters selected by a mask (Mallya & Lazebnik, 2018; Serra et al., 2018). In RRR we explored the addition of explanations to replay buffer and showed that saliency-based explanations offer performance upgrade as well as improvement in explanations across all memory-based and regularization-based baselines we tried.

**Visual explanation approaches** or saliency methods attempt to produce a posterior explanation or a pseudo-probability map for the detected signals from the target object in the input image. These approaches can be broadly divided into three categories including activation, gradient, and perturbation based methods. Activation-based methods (Zhou et al., 2016; Selvaraju et al., 2017; Chattopadhay et al., 2018) use a weighted linear combination of feature maps whereas gradient-based methods (Baehrens et al., 2010; Sundararajan et al., 2017a; Springenberg et al., 2014; Shrikumar et al., 2017; Zhang et al., 2018) use the derivative of outputs w.r.t the input image to compute pixel-wise importance scores to generate attention maps. Methods in these categories are only applicable to differentiable models, including convolutional neural networks (CNNs). In contrast, perturbation-based methods are model-agnostic and produce saliency maps by observing the change in prediction when the input is perturbed (Petsiuk et al., 2018; Ribeiro et al., 2016; Ross et al., 2017; Zhou et al., 2014; Seo et al., 2018). While these methods attempt to identify if models are right for the wrong reason, Ross et al. (2017) took a step further and applied penalties to correct the explanations provided in supervised/unsupervised fashion during training. Selvaraju et al. (2019) used human explanations in the form of question and answering to bring model explanations closer to human answers.

## 7 CONCLUSION

In this paper, we proposed the use of model explanations with continual learning algorithms to enhance better knowledge transfer as well as better recall of the previous tasks. The intuition behind our method is that encouraging a model to remember its evidence will increase the generalisability and rationality of recalled predictions and help retrieving the relevant aspects of each task. We advocate for the use of explainable AI as a tool to *improve* model performance, rather than as an artifact or interpretation of the model itself. We demonstrate that models which incorporate a "remember for the right reasons" constraint as part of a continual learning process can both be interpretable and more accurate. We empirically demonstrated the effectiveness of our approach in a variety of settings and provided an analysis of improved performance and explainability.

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

## A    APPENDIX

### A.1    GRAD-CAM TARGET LAYERS

Table 2 shows the target layer names used in Grad-CAM for different network architectures according to their standard PyTorch (Paszke et al., 2017) implementations. Saliency map size is equal to the activation map of the target layers.

Table 2: Target layer names and activation maps size for saliencies generated by different network architectures in Grad-CAM.

|  | Target layer name in PyTorch torchvision package | Saliency map size |
|---|---|---|
| SqueezeNet1_1 | `features.0.12.expand3x3` | $13 \times 13$ |
| AlexNet | `features.0.10` | $13 \times 13$ |
| ResNet18 | `features.7.1.conv2` | $7 \times 7$ |

## B    POINTING GAME VISUALIZATION

Figure 5 shows an example from CUB200 in the Pointing Game. We used the segmentation annotation to verify whether the peak of the predicted saliency map (marked with red cross) falls inside the object region. It is regarded as *hit* as the red cross is inside the segmentation mask for the bird.

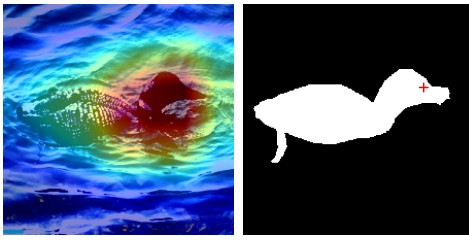

Figure 5: An example of PG evaluation as *hit* for an image in CUB200. Left: image saliency map overlaid on the image. Right: the segmentation label where the red cross marks the peak saliency.

## C    TABULAR RESULTS

In this section, we have tabulated results shown in Figure 2 and Figure 3 with means and standard deviations averaged over 3 runs.

Table 3: Classification accuracy of few-shot CIL learning of CUB200 at the end of 11 tasks for ER with and without $\mathcal{L}_{\text{RRR}}$ using different backbone architectures and saliency map techniques. Results are averaged over 3 runs. Figure 2 (left) in the main paper is generated using numbers in this Table.

|  | 1 | 2 | 3 | 4 | 5 | 6 | 7 | 8 | 9 | 10 | 11 |
|---|---|---|---|---|---|---|---|---|---|---|---|
| RN18-RRR-GCam | $67.8 \pm 0.8$ | $53.5 \pm 0.7$ | $45.6 \pm 0.6$ | $39.6 \pm 0.7$ | $35.3 \pm 0.9$ | $32.3 \pm 1.1$ | $29.4 \pm 0.9$ | $25.9 \pm 0.8$ | $25.7 \pm 0.6$ | $26.3 \pm 0.7$ | $23.6 \pm 0.7$ |
| RN18-ER | $67.8 \pm 0.8$ | $49.7 \pm 0.9$ | $41.7 \pm 0.8$ | $35.8 \pm 0.7$ | $31.4 \pm 0.9$ | $28.5 \pm 0.8$ | $25.5 \pm 0.8$ | $22.1 \pm 0.8$ | $21.8 \pm 0.8$ | $22.5 \pm 1.1$ | $19.8 \pm 0.9$ |
| RN18-RRR-Smooth | $67.8 \pm 0.8$ | $50.9 \pm 0.6$ | $43.5 \pm 0.9$ | $37.0 \pm 0.8$ | $33.0 \pm 0.7$ | $29.5 \pm 0.6$ | $26.8 \pm 0.8$ | $23.9 \pm 0.8$ | $23.9 \pm 0.8$ | $23.4 \pm 0.8$ | $21.5 \pm 0.5$ |
| RN18-RRR-BP | $67.8 \pm 0.8$ | $50.8 \pm 0.8$ | $43.9 \pm 0.6$ | $36.6 \pm 0.4$ | $32.7 \pm 0.6$ | $28.9 \pm 0.6$ | $27.2 \pm 0.5$ | $23.8 \pm 0.6$ | $23.8 \pm 0.6$ | $24.0 \pm 0.4$ | $21.5 \pm 0.6$ |
| RN18-Finetune | $67.8 \pm 0.8$ | $44.8 \pm 0.6$ | $32.2 \pm 0.5$ | $25.8 \pm 0.7$ | $25.6 \pm 0.7$ | $25.2 \pm 0.7$ | $20.8 \pm 0.6$ | $16.8 \pm 0.7$ | $18.8 \pm 0.5$ | $18.3 \pm 0.4$ | $17.1 \pm 0.6$ |
| Alex-RRR-GCam | $56.7 \pm 0.7$ | $46.6 \pm 0.5$ | $43.9 \pm 0.7$ | $41.3 \pm 0.7$ | $33.7 \pm 0.5$ | $27.4 \pm 0.7$ | $25.3 \pm 0.7$ | $22.0 \pm 0.5$ | $21.5 \pm 0.6$ | $21.4 \pm 0.6$ | $21.2 \pm 0.6$ |
| Alex-ER | $56.7 \pm 0.7$ | $44.6 \pm 0.7$ | $41.3 \pm 0.7$ | $38.7 \pm 0.7$ | $31.1 \pm 0.7$ | $24.5 \pm 0.7$ | $22.6 \pm 0.7$ | $19.6 \pm 0.6$ | $19.1 \pm 0.8$ | $18.7 \pm 0.8$ | $19.1 \pm 0.8$ |
| Alex-Finetune | $56.7 \pm 0.7$ | $42.8 \pm 0.8$ | $39.6 \pm 0.8$ | $36.9 \pm 0.8$ | $29.5 \pm 0.7$ | $23.3 \pm 0.6$ | $21.4 \pm 0.8$ | $17.9 \pm 0.7$ | $18.0 \pm 0.7$ | $17.0 \pm 0.5$ | $16.9 \pm 0.4$ |
| SQ-RRR-GCam | $46.8 \pm 0.5$ | $36.2 \pm 0.4$ | $30.1 \pm 0.6$ | $28.3 \pm 0.4$ | $25.1 \pm 0.5$ | $23.4 \pm 0.5$ | $19.3 \pm 0.6$ | $19.0 \pm 0.6$ | $18.5 \pm 0.5$ | $18.4 \pm 0.5$ | $18.2 \pm 0.6$ |
| SQ-ER | $46.8 \pm 0.5$ | $33.2 \pm 0.5$ | $27.1 \pm 0.6$ | $25.3 \pm 0.6$ | $22.1 \pm 0.5$ | $20.5 \pm 0.5$ | $16.3 \pm 0.4$ | $16.0 \pm 0.6$ | $15.5 \pm 0.6$ | $15.4 \pm 0.6$ | $15.2 \pm 0.7$ |
| SQ-Finetune | $46.8 \pm 0.5$ | $32.0 \pm 0.7$ | $25.2 \pm 0.7$ | $23.9 \pm 0.7$ | $20.2 \pm 0.8$ | $19.4 \pm 0.4$ | $14.9 \pm 0.4$ | $14.4 \pm 0.5$ | $13.8 \pm 0.4$ | $14.2 \pm 0.5$ | $13.7 \pm 0.6$ |

Table 4: Performance of the state-of-the-art existing approaches with and without $\mathcal{L}_{\text{RRR}}$ on CUB200 including TOPIC (Tao et al., 2020), EEIL (Castro et al., 2018), iCaRL (Rebuffi et al., 2017). Results for baselines are obtained using their original implementation. Results are averaged over 3 runs. Figure 2 (right) in the main paper is generated using numbers in this Table.

| | 1 | 2 | 3 | 4 | 5 | 6 | 7 | 8 | 9 | 10 | 11 |
|---|---|---|---|---|---|---|---|---|---|---|---|
| EEIL | $68.6 \pm 0.4$ | $53.6 \pm 0.4$ | $47.9 \pm 0.3$ | $44.2 \pm 0.8$ | $36.3 \pm 0.9$ | $27.4 \pm 1.2$ | $25.9 \pm 0.7$ | $24.7 \pm 0.5$ | $23.9 \pm 0.7$ | $24.1 \pm 0.7$ | $22.1 \pm 0.5$ |
| EEIL+RRR | $68.6 \pm 0.4$ | $56.6 \pm 0.5$ | $50.9 \pm 0.6$ | $48.3 \pm 0.5$ | $39.7 \pm 1.2$ | $31.4 \pm 0.7$ | $28.3 \pm 1.2$ | $28.0 \pm 0.6$ | $26.5 \pm 0.6$ | $27.4 \pm 0.6$ | $25.2 \pm 0.9$ |
| iCaRL | $68.6 \pm 0.4$ | $52.6 \pm 0.7$ | $48.6 \pm 1.2$ | $44.1 \pm 0.5$ | $36.6 \pm 0.3$ | $29.5 \pm 0.9$ | $27.8 \pm 0.4$ | $26.2 \pm 0.5$ | $24.0 \pm 0.6$ | $23.8 \pm 0.6$ | $21.1 \pm 0.7$ |
| iCaRL+RRR | $68.6 \pm 0.4$ | $55.6 \pm 1.2$ | $53.6 \pm 0.7$ | $47.1 \pm 0.8$ | $39.6 \pm 0.5$ | $32.5 \pm 0.8$ | $31.8 \pm 0.4$ | $29.2 \pm 0.6$ | $27.0 \pm 0.8$ | $27.8 \pm 0.6$ | $24.1 \pm 0.3$ |
| TOPIC | $68.6 \pm 0.4$ | $62.4 \pm 0.8$ | $54.8 \pm 0.4$ | $49.9 \pm 1.2$ | $45.2 \pm 0.6$ | $41.4 \pm 0.3$ | $38.3 \pm 0.8$ | $35.3 \pm 0.6$ | $32.2 \pm 0.3$ | $28.3 \pm 0.6$ | $26.2 \pm 1.2$ |
| TOPIC+RRR | $68.6 \pm 0.4$ | $62.5 \pm 0.9$ | $56.8 \pm 0.4$ | $51.5 \pm 0.5$ | $48.2 \pm 0.4$ | $44.4 \pm 0.4$ | $42.3 \pm 0.7$ | $38.3 \pm 0.6$ | $35.2 \pm 0.9$ | $32.3 \pm 0.9$ | $29.2 \pm 0.5$ |
| FT | $68.6 \pm 0.4$ | $44.8 \pm 0.5$ | $32.2 \pm 0.8$ | $25.8 \pm 0.4$ | $25.6 \pm 1.1$ | $25.2 \pm 0.7$ | $20.8 \pm 1.1$ | $16.7 \pm 0.4$ | $18.8 \pm 1.1$ | $18.2 \pm 0.3$ | $17.1 \pm 0.8$ |
| ER | $67.8 \pm 0.8$ | $49.7 \pm 0.9$ | $41.7 \pm 0.8$ | $35.8 \pm 0.7$ | $31.4 \pm 0.9$ | $28.5 \pm 0.8$ | $25.5 \pm 0.8$ | $22.1 \pm 0.8$ | $21.8 \pm 0.6$ | $22.5 \pm 1.1$ | $19.8 \pm 0.9$ |
| RRR | $67.8 \pm 0.8$ | $53.5 \pm 0.7$ | $45.6 \pm 0.6$ | $39.6 \pm 0.7$ | $35.3 \pm 0.9$ | $32.3 \pm 1.1$ | $29.4 \pm 0.9$ | $25.9 \pm 0.8$ | $25.7 \pm 0.6$ | $26.3 \pm 0.7$ | $23.6 \pm 0.7$ |
| JT | $68.6 \pm 0.4$ | $62.4 \pm 0.4$ | $57.2 \pm 0.4$ | $52.8 \pm 0.5$ | $49.5 \pm 0.9$ | $46.1 \pm 0.5$ | $42.8 \pm 1.1$ | $40.1 \pm 0.8$ | $38.7 \pm 0.7$ | $37.1 \pm 0.5$ | $35.6 \pm 0.9$ |

Table 5: Performance of the state-of-the-art existing approaches with and without $\mathcal{L}_{\text{RRR}}$ on CI-FAR100 in 10 tasks. Results for iTAML (Rajasegaran et al., 2020), BiC (Wu et al., 2019), and EEIL (Castro et al., 2018) are produced with their original implementation while EWC (Kirkpatrick et al., 2017) and LwF (Li & Hoiem, 2016) are re-implemented by us. Results are averaged over 3 runs. Figure 3a in the main paper is generated using numbers in this Table.

| | 1 | 2 | 3 | 4 | 5 | 6 | 7 | 8 | 9 | 10 |
|---|---|---|---|---|---|---|---|---|---|---|
| iTAML+RRR | $89.2 \pm 0.5$ | $92.3 \pm 0.7$ | $89.5 \pm 1.2$ | $87.5 \pm 1.2$ | $84.1 \pm 0.8$ | $83.5 \pm 0.9$ | $83.9 \pm 0.7$ | $81.2 \pm 0.3$ | $79.6 \pm 0.9$ | $79.7 \pm 0.5$ |
| iTAML | $89.2 \pm 0.5$ | $88.9 \pm 0.5$ | $87.0 \pm 1.1$ | $85.7 \pm 1.1$ | $84.1 \pm 1.1$ | $81.8 \pm 0.3$ | $80.0 \pm 0.6$ | $79.0 \pm 0.3$ | $78.6 \pm 0.8$ | $77.8 \pm 0.6$ |
| BiC | $90.3 \pm 0.7$ | $82.1 \pm 0.7$ | $75.1 \pm 0.4$ | $69.8 \pm 1.2$ | $65.3 \pm 0.8$ | $61.3 \pm 0.9$ | $57.4 \pm 0.7$ | $54.9 \pm 0.5$ | $53.2 \pm 0.9$ | $50.3 \pm 0.7$ |
| BiC+RRR | $90.3 \pm 0.7$ | $84.9 \pm 1.1$ | $76.4 \pm 0.6$ | $69.3 \pm 0.3$ | $65.1 \pm 0.9$ | $63.3 \pm 0.4$ | $59.7 \pm 1.1$ | $55.4 \pm 0.8$ | $55.8 \pm 0.7$ | $52.1 \pm 0.5$ |
| EEIL | $80.0 \pm 0.7$ | $80.5 \pm 1.2$ | $75.5 \pm 0.9$ | $71.5 \pm 0.4$ | $68.0 \pm 1.2$ | $62.0 \pm 0.9$ | $59.0 \pm 0.7$ | $55.1 \pm 1.2$ | $51.4 \pm 0.8$ | $48.7 \pm 0.4$ |
| EEIL+RRR | $80.0 \pm 0.7$ | $83.5 \pm 0.3$ | $78.7 \pm 1.2$ | $74.0 \pm 1.2$ | $71.7 \pm 0.3$ | $65.1 \pm 0.4$ | $61.2 \pm 0.5$ | $57.6 \pm 0.5$ | $54.1 \pm 0.4$ | $51.7 \pm 0.3$ |
| LwF | $86.1 \pm 1.2$ | $69.0 \pm 0.7$ | $55.0 \pm 0.3$ | $45.8 \pm 0.3$ | $40.4 \pm 0.5$ | $36.7 \pm 0.9$ | $30.8 \pm 0.7$ | $28.6 \pm 0.5$ | $26.1 \pm 0.7$ | $24.2 \pm 0.7$ |
| LwF+RRR | $86.1 \pm 1.2$ | $72.4 \pm 0.8$ | $57.0 \pm 1.1$ | $48.3 \pm 0.3$ | $43.2 \pm 0.8$ | $39.3 \pm 0.5$ | $34.1 \pm 0.6$ | $32.1 \pm 1.1$ | $29.8 \pm 0.7$ | $27.1 \pm 0.6$ |
| EWC | $86.1 \pm 1.2$ | $52.6 \pm 0.4$ | $48.6 \pm 0.4$ | $38.5 \pm 0.5$ | $31.1 \pm 0.9$ | $26.5 \pm 0.3$ | $21.7 \pm 0.6$ | $20.0 \pm 0.7$ | $18.9 \pm 0.5$ | $16.6 \pm 0.9$ |
| EWC+RRR | $86.1 \pm 1.2$ | $56.0 \pm 0.4$ | $53.9 \pm 1.2$ | $44.4 \pm 0.9$ | $35.1 \pm 0.5$ | $28.6 \pm 0.6$ | $25.1 \pm 1.1$ | $23.1 \pm 0.5$ | $18.8 \pm 0.9$ | $19.0 \pm 1.2$ |
| ER | $86.1 \pm 1.2$ | $74.5 \pm 0.9$ | $65.2 \pm 0.8$ | $62.5 \pm 0.8$ | $56.7 \pm 0.7$ | $50.5 \pm 0.3$ | $47.6 \pm 0.4$ | $43.4 \pm 0.3$ | $41.6 \pm 0.9$ | $38.1 \pm 1.1$ |
| RRR | $86.1 \pm 1.2$ | $78.5 \pm 0.9$ | $69.2 \pm 1.1$ | $63.5 \pm 1.2$ | $58.7 \pm 0.8$ | $53.5 \pm 1.1$ | $49.6 \pm 0.7$ | $44.4 \pm 0.3$ | $42.6 \pm 1.2$ | $39.1 \pm 1.1$ |

Table 7: Performance of the state-of-the-art existing approaches with and without $\mathcal{L}_{\text{RRR}}$ on ImageNet100 in 10 tasks. Results for iTAML (Rajasegaran et al., 2020), BiC (Wu et al., 2019), and EEIL (Castro et al., 2018) are produced with their original implementation while EWC (Kirkpatrick et al., 2017) and LwF (Li & Hoiem, 2016) are re-implemented by us. Results are averaged over 3 runs. Figure 3c in the main paper is generated using numbers in this Table.

| | 1 | 2 | 3 | 4 | 5 | 6 | 7 | 8 | 9 | 10 |
|---|---|---|---|---|---|---|---|---|---|---|
| iTAML | $99.4 \pm 0.8$ | $96.4 \pm 0.9$ | $94.4 \pm 0.9$ | $93.0 \pm 0.3$ | $92.4 \pm 1.2$ | $90.6 \pm 0.3$ | $89.9 \pm 0.4$ | $90.3 \pm 0.8$ | $90.3 \pm 1.1$ | $89.8 \pm 0.4$ |
| iTAML+RRR | $99.4 \pm 0.8$ | $97.3 \pm 0.5$ | $96.6 \pm 0.7$ | $96.3 \pm 1.1$ | $95.3 \pm 0.5$ | $93.1 \pm 0.5$ | $92.1 \pm 0.6$ | $92.1 \pm 0.6$ | $92.9 \pm 0.9$ | $91.9 \pm 0.4$ |
| EEIL | $99.5 \pm 0.4$ | $98.8 \pm 1.1$ | $95.9 \pm 0.9$ | $93.0 \pm 0.4$ | $88.3 \pm 1.1$ | $86.7 \pm 1.2$ | $83.0 \pm 1.2$ | $81.1 \pm 0.5$ | $78.2 \pm 0.7$ | $75.4 \pm 0.4$ |
| EEIL+RRR | $99.5 \pm 0.4$ | $98.1 \pm 0.7$ | $97.4 \pm 1.1$ | $96.7 \pm 0.4$ | $93.3 \pm 0.5$ | $89.4 \pm 1.1$ | $86.5 \pm 0.3$ | $86.1 \pm 1.1$ | $81.8 \pm 0.4$ | $77.0 \pm 0.3$ |
| BiC | $98.3 \pm 0.7$ | $94.9 \pm 0.8$ | $93.5 \pm 0.7$ | $90.9 \pm 1.2$ | $89.0 \pm 1.2$ | $87.3 \pm 0.6$ | $85.2 \pm 0.7$ | $83.2 \pm 0.4$ | $82.5 \pm 0.9$ | $81.1 \pm 1.1$ |
| BiC+RRR | $98.3 \pm 0.7$ | $98.9 \pm 0.3$ | $96.5 \pm 0.6$ | $93.9 \pm 0.4$ | $92.0 \pm 0.7$ | $89.3 \pm 1.1$ | $87.2 \pm 0.8$ | $87.2 \pm 1.1$ | $85.5 \pm 0.9$ | $84.1 \pm 0.6$ |
| iCaRL | $99.3 \pm 0.4$ | $97.2 \pm 0.9$ | $93.5 \pm 0.9$ | $91.0 \pm 0.3$ | $87.5 \pm 1.2$ | $82.1 \pm 1.2$ | $77.1 \pm 0.4$ | $72.8 \pm 0.6$ | $67.1 \pm 0.8$ | $63.5 \pm 1.1$ |
| iCaRL+RRR | $99.3 \pm 0.4$ | $97.9 \pm 1.2$ | $94.1 \pm 0.7$ | $92.8 \pm 0.7$ | $91.7 \pm 0.9$ | $85.7 \pm 1.1$ | $82.1 \pm 0.6$ | $74.4 \pm 0.9$ | $72.2 \pm 0.8$ | $68.1 \pm 0.9$ |
| LwF | $99.3 \pm 0.5$ | $95.2 \pm 0.9$ | $85.9 \pm 0.9$ | $73.9 \pm 1.1$ | $63.7 \pm 0.8$ | $54.8 \pm 0.8$ | $50.1 \pm 0.6$ | $44.5 \pm 0.9$ | $40.7 \pm 0.5$ | $36.7 \pm 0.3$ |
| LwF+RRR | $99.3 \pm 0.5$ | $97.1 \pm 1.2$ | $89.3 \pm 0.6$ | $79.1 \pm 0.5$ | $69.1 \pm 1.1$ | $59.4 \pm 1.1$ | $57.2 \pm 0.7$ | $48.2 \pm 1.1$ | $45.1 \pm 0.6$ | $41.5 \pm 0.5$ |
| FT | $99.3 \pm 0.5$ | $49.4 \pm 0.3$ | $32.6 \pm 0.3$ | $24.7 \pm 0.6$ | $20.0 \pm 1.2$ | $16.7 \pm 0.3$ | $13.9 \pm 0.3$ | $12.3 \pm 0.7$ | $11.1 \pm 0.6$ | $9.9 \pm 0.7$ |
| ER | $99.3 \pm 0.5$ | $95.2 \pm 0.8$ | $88.1 \pm 0.8$ | $78.1 \pm 0.9$ | $72.5 \pm 0.6$ | $69.1 \pm 0.8$ | $67.1 \pm 0.6$ | $61.8 \pm 0.6$ | $55.1 \pm 0.3$ | $50.1 \pm 1.1$ |
| RRR | $99.3 \pm 0.5$ | $96.5 \pm 0.3$ | $93.4 \pm 0.8$ | $84.8 \pm 0.7$ | $78.7 \pm 0.4$ | $74.7 \pm 0.4$ | $73.1 \pm 0.5$ | $68.4 \pm 0.8$ | $60.2 \pm 0.3$ | $55.1 \pm 0.7$ |

Table 6: Performance of the state-of-the-art existing approaches with and without $\mathcal{L}_{\text{RRR}}$ on CI-FAR100 in 20 tasks. Results for iTAML (Rajasegaran et al., 2020), BiC (Wu et al., 2019), and EEIL (Castro et al., 2018) are produced with their original implementation while EWC (Kirkpatrick et al., 2017) and LwF (Li & Hoiem, 2016) are re-implemented by us. Results are averaged over 3 runs. Figure 3b in the main paper is generated using numbers in this Table.

(a) Tasks 1-10

|  | 1 | 2 | 3 | 4 | 5 | 6 | 7 | 8 | 9 | 10 |
|---|---|---|---|---|---|---|---|---|---|---|
| iTAML | $84.7 \pm 0.6$ | $85.7 \pm 0.4$ | $86.5 \pm 0.3$ | $86.5 \pm 0.8$ | $86.3 \pm 1.2$ | $85.7 \pm 0.8$ | $84.9 \pm 1.1$ | $82.6 \pm 0.3$ | $80.8 \pm 0.7$ | $82.4 \pm 0.3$ |
| iTAML+RRR | $84.7 \pm 0.6$ | $89.9 \pm 0.5$ | $89.2 \pm 0.9$ | $89.2 \pm 0.6$ | $89.0 \pm 1.1$ | $87.2 \pm 0.6$ | $88.0 \pm 0.4$ | $85.6 \pm 1.1$ | $86.6 \pm 0.3$ | $85.4 \pm 0.3$ |
| BiC | $95.7 \pm 0.6$ | $90.3 \pm 0.9$ | $80.9 \pm 0.8$ | $75.8 \pm 0.8$ | $73.5 \pm 0.6$ | $71.5 \pm 1.2$ | $67.8 \pm 0.4$ | $65.4 \pm 0.8$ | $62.7 \pm 1.2$ | $61.9 \pm 1.2$ |
| BiC+RRR | $95.7 \pm 0.6$ | $93.3 \pm 0.6$ | $84.7 \pm 1.1$ | $77.5 \pm 0.9$ | $73.4 \pm 0.6$ | $74.8 \pm 0.6$ | $69.6 \pm 0.7$ | $67.4 \pm 0.3$ | $65.7 \pm 0.5$ | $64.9 \pm 0.6$ |
| EEIL | $81.9 \pm 0.5$ | $86.3 \pm 0.3$ | $84.9 \pm 0.4$ | $80.7 \pm 0.3$ | $77.7 \pm 0.6$ | $74.9 \pm 0.3$ | $70.9 \pm 0.7$ | $67.4 \pm 0.7$ | $64.9 \pm 0.5$ | $62.4 \pm 0.3$ |
| EEIL+RRR | $81.9 \pm 0.5$ | $88.4 \pm 0.8$ | $87.6 \pm 0.7$ | $82.6 \pm 1.2$ | $78.5 \pm 0.6$ | $76.9 \pm 0.4$ | $71.2 \pm 0.7$ | $67.3 \pm 0.4$ | $67.0 \pm 1.2$ | $64.5 \pm 0.3$ |
| LwF | $85.1 \pm 0.7$ | $68.8 \pm 0.9$ | $58.6 \pm 1.1$ | $50.5 \pm 1.2$ | $43.5 \pm 0.9$ | $37.5 \pm 0.6$ | $33.7 \pm 0.9$ | $30.4 \pm 0.9$ | $26.8 \pm 1.1$ | $24.9 \pm 0.7$ |
| LwF+RRR | $85.1 \pm 0.7$ | $71.6 \pm 0.6$ | $61.8 \pm 0.7$ | $54.2 \pm 0.5$ | $46.2 \pm 0.9$ | $40.7 \pm 0.7$ | $36.7 \pm 1.2$ | $34.4 \pm 0.4$ | $29.8 \pm 0.7$ | $27.2 \pm 1.2$ |
| EWC | $85.1 \pm 0.7$ | $61.3 \pm 0.5$ | $47.4 \pm 0.8$ | $36.2 \pm 0.3$ | $31.3 \pm 0.6$ | $27.9 \pm 0.5$ | $23.7 \pm 1.1$ | $22.5 \pm 0.4$ | $20.8 \pm 0.8$ | $18.9 \pm 0.7$ |
| EWC+RRR | $85.1 \pm 0.7$ | $68.9 \pm 0.5$ | $52.2 \pm 0.9$ | $39.9 \pm 0.9$ | $35.2 \pm 0.3$ | $30.0 \pm 0.3$ | $24.3 \pm 0.8$ | $24.0 \pm 0.6$ | $23.7 \pm 0.4$ | $21.0 \pm 1.1$ |
| ER | $85.1 \pm 0.7$ | $83.1 \pm 0.9$ | $81.8 \pm 0.7$ | $74.9 \pm 0.3$ | $70.4 \pm 0.3$ | $61.5 \pm 1.2$ | $60.8 \pm 1.1$ | $57.0 \pm 0.7$ | $54.3 \pm 0.4$ | $48.2 \pm 0.6$ |
| RRR | $85.1 \pm 0.7$ | $85.1 \pm 0.9$ | $83.8 \pm 0.4$ | $77.9 \pm 0.4$ | $72.4 \pm 1.2$ | $64.5 \pm 0.7$ | $62.8 \pm 0.7$ | $59.0 \pm 0.3$ | $57.3 \pm 0.8$ | $51.2 \pm 1.1$ |

(b) Tasks 11-20

|  | 11 | 12 | 13 | 14 | 15 | 16 | 17 | 18 | 19 | 20 |
|---|---|---|---|---|---|---|---|---|---|---|
| iTAML | $80.0 \pm 1.1$ | $80.6 \pm 0.5$ | $74.3 \pm 0.8$ | $70.7 \pm 0.6$ | $71.3 \pm 1.1$ | $68.3 \pm 0.5$ | $70.3 \pm 0.8$ | $68.3 \pm 0.6$ | $69.5 \pm 0.3$ | $66.0 \pm 0.6$ |
| iTAML+RRR | $85.5 \pm 0.5$ | $85.2 \pm 0.8$ | $79.7 \pm 0.6$ | $74.3 \pm 0.4$ | $74.0 \pm 0.9$ | $73.4 \pm 1.1$ | $74.8 \pm 0.9$ | $74.4 \pm 0.4$ | $73.9 \pm 0.5$ | $71.8 \pm 0.9$ |
| BiC | $59.2 \pm 0.4$ | $57.0 \pm 0.6$ | $56.1 \pm 1.2$ | $55.7 \pm 0.6$ | $53.8 \pm 0.5$ | $52.4 \pm 1.2$ | $49.7 \pm 0.6$ | $49.2 \pm 1.2$ | $47.7 \pm 1.1$ | $46.7 \pm 1.2$ |
| BiC+RRR | $62.2 \pm 0.5$ | $59.1 \pm 0.7$ | $58.2 \pm 0.5$ | $57.8 \pm 0.5$ | $54.4 \pm 1.2$ | $56.6 \pm 0.9$ | $53.9 \pm 0.7$ | $52.4 \pm 1.1$ | $49.5 \pm 0.8$ | $49.4 \pm 0.9$ |
| EEIL | $60.9 \pm 0.6$ | $59.5 \pm 0.6$ | $57.8 \pm 0.6$ | $55.1 \pm 0.3$ | $53.9 \pm 0.5$ | $51.7 \pm 0.3$ | $50.1 \pm 0.8$ | $49.4 \pm 0.5$ | $47.4 \pm 0.6$ | $46.9 \pm 0.9$ |
| EEIL+RRR | $63.7 \pm 0.6$ | $62.9 \pm 0.4$ | $59.7 \pm 0.4$ | $57.0 \pm 0.3$ | $55.6 \pm 0.8$ | $53.5 \pm 0.4$ | $53.5 \pm 0.3$ | $52.7 \pm 0.4$ | $49.1 \pm 0.3$ | $47.8 \pm 0.4$ |
| LwF | $23.9 \pm 0.7$ | $21.4 \pm 0.7$ | $20.0 \pm 0.7$ | $19.1 \pm 0.9$ | $18.7 \pm 0.8$ | $17.1 \pm 0.8$ | $15.6 \pm 0.8$ | $14.7 \pm 0.8$ | $14.0 \pm 0.8$ | $13.7 \pm 1.1$ |
| LwF+RRR | $27.7 \pm 0.7$ | $26.9 \pm 0.9$ | $25.7 \pm 0.7$ | $24.5 \pm 1.2$ | $23.6 \pm 0.6$ | $22.6 \pm 0.7$ | $19.5 \pm 0.3$ | $18.6 \pm 0.5$ | $19.7 \pm 0.8$ | $18.4 \pm 1.2$ |
| EWC | $17.2 \pm 1.1$ | $16.0 \pm 0.5$ | $15.0 \pm 0.8$ | $14.5 \pm 0.8$ | $13.4 \pm 1.1$ | $12.4 \pm 0.4$ | $12.3 \pm 0.4$ | $11.5 \pm 0.8$ | $11.2 \pm 0.8$ | $9.44 \pm 0.5$ |
| EWC+RRR | $20.7 \pm 0.3$ | $19.5 \pm 0.4$ | $18.4 \pm 0.7$ | $17.3 \pm 0.5$ | $16.2 \pm 0.4$ | $15.8 \pm 0.5$ | $15.0 \pm 0.7$ | $16.6 \pm 0.9$ | $14.3 \pm 0.4$ | $13.2 \pm 0.3$ |
| ER | $45.8 \pm 0.6$ | $42.7 \pm 0.7$ | $41.6 \pm 0.6$ | $41.2 \pm 0.6$ | $36.5 \pm 0.4$ | $36.5 \pm 0.6$ | $33.8 \pm 0.4$ | $32.4 \pm 1.2$ | $31.4 \pm 0.7$ | $30.2 \pm 0.5$ |
| RRR | $48.8 \pm 0.3$ | $46.7 \pm 0.9$ | $43.6 \pm 1.1$ | $44.2 \pm 0.7$ | $39.5 \pm 0.3$ | $38.5 \pm 0.9$ | $35.8 \pm 0.3$ | $33.4 \pm 0.3$ | $32.4 \pm 0.3$ | $31.2 \pm 0.3$ |

