# OpenReview forum: "Remembering for the Right Reasons: Explanations Reduce Catastrophic Forgetting"
_ICLR.cc/2021/Conference — ICLR 2021 Poster_

### Official Review · AnonReviewer2 · 2020-10-26
**I vote for acceptance as the paper brings a notable performance boost in continual learning with a simple yet meaningful objective.**

**Rating:** 6
**Confidence:** 4

**Review:**

**Summary**

This paper delivers an additional training objective for continual learning which regularizes the update of the model parameters by making them not deviate a lot from its temporal snapshots of the visual saliency map. The additional information, *reason*, that the regularization exploits helps to solve the problem of catastrophic forgetting in addition to providing a qualitative explanation.

**Strong points**

- The method, RRR, is simple to implement for most visual continual learning task setting, yet to yield a significant performance boost.
- The evaluation verifies its efficacy throughout many state-of-the-art continual learning methods.
- As a byproduct, the method provides a qualitative explanation map, which in most cases can only be acquired by an additional posthoc training process.

**Weak points**

- The method needs input to be spatially meaningful. Although most of the continual learning tasks currently deal only with the visual domain, in general, the constraints they have are only a condition that is a classification problem. Therefore the method presented in this paper will bring additional constraints to the input and can not be directly applicable when the input is not spatially meaningful (e.g., set classification).

**Recommendation**

I vote for acceptance as the paper brings a notable performance boost in continual learning with a simple yet meaningful objective.

**Supporting arguments for the recommendation**

In addition to the performance improvements described above, the method presented in this paper is expected to be of great help to the XAI community as well. Considering the impact of this paper on the XAI community that saliency maps and other explanation techniques (although they have to be differentiable w.r.t. parameter) can lead to such direct quantitative performance improvement beyond just XAI use, I think it is sufficient to accept this paper.

**Additional feedbacks**

As saliency maps are densities with fixed support (fixed image size), I think OT distances can be employed in place of L1 loss. Using L1 loss makes the model hard to discriminate the case when the prediction center of mass is far away from the target center versus the case when the prediction center is still wrong but close to the target center. For example, if target saliency is focused on the wings of a bird image, then predictions that one focusing on the bird's torso (which is very close to its wings) and the other focusing on the image corner equally penalized by L1 loss. Optimal transport distances can handle these cases and I think its employment can boost performance.

**Typos**

- The last paragraph of section 2 seems incomplete (~~~ pass what? pass upstream gradient?)
- $S$* is missing in the equation 2. maybe $\hat{s}$?

---

> ### Author Response · Authors · 2020-11-20
> **Responses to individual comments from AnonReviewer2**
>
> We would like to thank you for your time and feedback. Please see our responses to your questions in the orders they were asked. We will also upload the updated pdf shortly.
>
> 1. Your feedback regarding using L2 loss is a valuable point. We also expected to see the same behavior and tried L2 but achieved a slightly better results with L1 loss on CUB dataset and used L1 throughout the remaining experiments. Our takeaway is that because the memory samples are randomly selected, their predictions can be arbitrary and the saliency maps might look meaningful or might be as bad as predicting a uniform probability distribution over the entire mask which might cause generating some ‘outliers’ in our saliency maps training samples which is a situation that L2 loss is known to perform worse that L1. Having said that, it should be noted that RRR’s goal is to preserve the saliency maps, regardless of their quality or correctness, and hence is not affected by this artifact.
>
> 2. It appears that we commented a few words from this sentence by mistake. Here is the complete form with missing parts marked with bold font. We also updated the text in the paper:
>
> Unlike  the  common  saliency  map  techniques  of  Guided  BackProp  (Springenberg  et  al.,  2014), Guided  GradCAM  (Selvaraju  et  al.,  2016),  Integrated  Gradients  (Sundararajan  et  al.,  2017b), Gradient $\cdot$ Input  (Shrikumar  et  al.,  2016),  SmoothGrad  etc.,  **only**  vanilla  Backpropagation  and Grad-CAM pass **important “sanity checks” regarding their sensitivity to data and model parameters (Adebayo et al., 2018).**
>
> Adebayo, J., Gilmer, J., Muelly, M., Goodfellow, I., Hardt, M., & Kim, B. (2018). Sanity checks for saliency maps. In Advances in Neural Information Processing Systems (pp. 9505-9515).
>
> 3. We removed $S^*$ and changed it to $\hat{s}$.

---

### Official Review · AnonReviewer3 · 2020-10-28
**Review for Remembering for the Right Reasons: Explanations Reduce Catastrophic Forgetting**

**Rating:** 6
**Confidence:** 2

**Review:**

This paper proposes a method for continual learning of a sequence of supervised tasks which is based on memory-replay for remembering evidence from previously made decisions. This evidence relies on explanations of those decisions rather than data. These explanations are incorporated directly into the loss function. The proposed approach is tested against several counterparts using benchmark datasets in continual learning.

I would like to remark as strengths of the paper:
-	The method adopts a novel approach of incorporating explanations along with data for
-	The experiments provide clear insights onto which kinds of explanations would work better, by exploring several strategies like vanilla backpropagation, Grad-CAM, etc.
-	The paper provides a thoughtful analysis of the impact of explanations that demonstrates some advantages in terms of accuracy and backward transfer.

Some weaknesses of the paper are:
-	As the examples and explanations to remember are selected randomly in the experiments, it is perhaps a good idea to perform this process several times to guarantee reliability of these results. Furthermore, does the number of examples and explanations depend on the dataset size, i.e. is there a relation between the number of training examples, the memory replay examples and the performance of the proposed method?

Other comments:
-	I suggest to improve clarity of the paper by improving some of the language used, e.g. in page 4 it’s said that “We show below that combining RRR into the objective function of state-of-the-art memory and regularization-based methods results in significant performance improvements.”. In this case, “significant” is clearly not the word to be used as the paper does not include statistical significance tests supporting this claim!

Questions for authors:
- Please address questions in the comments above.

---

> ### Author Response · Authors · 2020-11-20
> **Responses to individual comments from AnonReviewer3**
>
> We would like to thank you for your time and feedback. Please see our responses to your questions in the orders they were asked. We will also upload the updated pdf shortly.
>
> 1. All of our results are averaged over 3 runs. We added ables to our supplementary materials to show the average numbers and standard deviations. We have used the common memory size used for the benchmarks to be consistent with the literature. This approach, similar to all the regularization-based approaches in continual learning, is limited to the number of tasks it can scale to as they all have a fixed network architecture which does not change in size. Hence choosing a large memory budget for $M^{RRR}$ would potentially prevent the model from learning new tasks. We added this limitation to the end of Section 3.2 in the main text.
>
> 2. We agree with the reviewer and remove the word significant from the text.

---

### Official Review · AnonReviewer1 · 2020-10-28
**This proposed technique is very simple. It saves reasons for classification and use them in continual learning in addition to saved examples.**

**Rating:** 6
**Confidence:** 4

**Review:**

The proposed technique is simple. It helps existing replay methods to boost their accuracy for class-incremental learning. In addition to save some training examples of previous tasks or classes, it also saves the saliency maps of these examples as explanations of classification. The size the saliency maps is small and requires very little memory. For continual learning, it includes additional term in the loss function to take care of the saved saliency maps. Experimental results show that this additional saved information can help improve existing replay methods for class-incremental learning.

Can you give some reasons why the proposed method works? Replaying previous examples may help continual learning in two ways, preserving the previous parameters and/or correcting previous parameters/features if they are insufficient to discriminate old classes in old tasks and the new classes in the new task. The saved saliency maps mainly play/enhance the first role, but that may conflict with the second role? If that is true, the saliency maps may help the continual learning performance in some cases but harm it in some other cases? Can you try CIFAR10 and MNIST with 2 classes in each task? When you have fewer classes in a task, the second role may be more important. Also, since you have tried CIFAR100 with 10 and 20 tasks, can you also try ImageNet100 with 20 tasks?

Since the memory required by the Grad-CAM saliency maps are small, it is possible to save more of them without using any saved training images?

Can your technique help those methods that don’t use replay?

I thought M^{RRR} is the model explanation memory. What is xai buffer memory M^{xai}? Are they the same?

I prefer tables to line plots when showing results as it is hard to see how much the improvement is in line plots. The three sub-figures are also too small to see.

I have difficulty to parse the first sentence in the last paragraph of section 2. Anything wrong there?

---

> ### Author Response · Authors · 2020-11-20
> **Responses to individual comments from AnonReviewer1**
>
> We would like to thank you for your time and feedback. Please see our responses to your questions in the orders they were asked. We will also upload the updated pdf shortly.
> 1. The stored saliency maps are a linear combination of feature maps for a relatively small memory size that we try to preserve for those samples assuming that they represent the entire old task data. We think that having too many samples stored for prior tasks would apply more restriction on the feature maps and will prevent the parameters from being changed in favor of new tasks. So we agree with the reviewer that it can potentially harm the performance if $M^{RRR}$ is too large.
> Here are the results for the requested experiments. In MNIST and CIFAR10 with two tasks, we only see a gain of 0.1% and 0.3% by using RRR over ER, respectively. In ImageNet experiment with $20$ tasks, RRR improves ER but the difference between the accuracies decreases as the number of tasks grows.
>
> **MNIST** (average of 3 runs)
>
> **RRR:**  [$98.5\pm0.04$, $99.7\pm0.03$]
>
> **ER:**    [$98.5\pm0.04$, $99.6\pm0.02$]
>
>
> **CIFAR10** (average of 3 runs)
>
> **RRR:**  [$89.2\pm0.1$, $88.1\pm0.2$]
>
> **ER:**    [$89.2\pm0.1$, $87.8\pm0.2$]
>
> **ImageNet100 in 20 tasks:** (one run only):
>
> **RRR:** [99.1 ,98.3, 96.2 , 86.8, 84.7 ,81.7, 79.2 , 77.4, 73.6, 71.7, 68.4, 63.7, 59.0, 56.1,  54.1 ,50.2  , 44.5   ,43.6 , 43.2 , 41.2 , 40.4]
>
> **ER:**  [99.1 ,97.1, 95.1 , 83.4, 81.4 ,79.2, 75.2 , 75.1, 70.0, 68.3, 65.1, 61.1, 57.2, 54.3,  53.2 ,49.6  , 43.6   ,43.0 , 42.8 , 40.8 , 39.8]
>
> 2. Yes, you can remove the old images from the training data and only use them to encourage the model to produce the same saliency map for them as it learns new tasks. We did this for LwF and EWC experiments on CIFAR100 and ImageNet100 as those are regularization-based methods and do not use experience replay. The accuracy improved for them despite not benefiting from revisiting the raw data. Results for this are already shown in Figure 3 and explained in Section 4.2.
>
> 3. Yes, we have shown results in Figure 3 for adding RRR to EWC and LwF on CIFAR100 and ImageNet which are both regularization-based only methods and do not use experience replay and showed improvement in their performance versus without using RRR.
>
> 4. Yes,  $M^{xai}$ and  $M^{RRR}$ are the same. We changed  $M^{xai}$ with $M^{RRR}$ that on the third paragraph of page 2.
>
> 5. We added tables to our supplementary material to show the numbers used to generate all of our plots.
>
> 6. It appears that we commented a few words from this sentence by mistake. Here is the complete form with missing parts marked with bold font. We also updated the text in the paper:
>
> Unlike  the  common  saliency  map  techniques  of  Guided  BackProp  (Springenberg  et  al.,  2014), Guided  GradCAM  (Selvaraju  et  al.,  2016),  Integrated  Gradients  (Sundararajan  et  al.,  2017b), Gradient $\cdot$ Input  (Shrikumar  et  al.,  2016),  SmoothGrad  etc.,  **only**  vanilla  Backpropagation  and Grad-CAM pass **important “sanity checks” regarding their sensitivity to data and model parameters (Adebayo et al., 2018).**
>
> Adebayo, J., Gilmer, J., Muelly, M., Goodfellow, I., Hardt, M., & Kim, B. (2018). Sanity checks for saliency maps. In Advances in Neural Information Processing Systems (pp. 9505-9515).

---

### Official Review · AnonReviewer4 · 2020-10-28
**A simple/effective distillation-based approach to ease catastrophic forgetting for continual learning**

**Rating:** 6
**Confidence:** 4

**Review:**

Summary
-------------

This paper tackles the problem of catastrophic forgetting in a continual learning scenario, in which the same classifier is trained incrementally on new classification tasks, each defined on a new set of output classes, and asked to retain performance on all the previous tasks. To tackle this problem, a stream of approaches keep a minimal "replay buffer" of examples and their labels from each task, that can be accessed during sequential task learning. This paper belongs to this stream of approaches. A naive way of using the replay buffer is to mix the buffer to the current task training set. Additionally, this paper proposes to store gradient attribution maps in the replay buffer and use the stored gradient attribution maps as additional targets for the current model (RRR loss): the model should have "explanations" of previous examples that roughly don't change upon seeing new evidence from other classes. The authors show that the proposed RRR loss can be applied to multiple baselines and improves performance on a continual learning and few-shot continual learning scenario.

The main contribution of the paper is to use and store gradient attribution maps along with the examples and use the attribution maps as additional targets for remembering previous tasks. This goes into the direction of extracting "summaries" of previous models that can be re-used to inform the current model and avoid forgetting. The paper is well-written and clear. I particularly like the simplicity of approach of the paper and I think the results support its effectiveness. However, in the current form, I cannot quite conclude that it's the "explanations" that are really crucial for this to work and not potentially other statistics that can be extracted from the model. Also, it's unclear what are the negative effects of the proposed approach for backward transfer. About this, some questions / suggestion below.

Pros
------
- Approach is simple, quite general and improves on a variety of baselines

Cons
-------
- Is it really the "explanation" that is important ? A naive baseline is missing (details afterwards)
- Conceptually, the limits of this approach for backward transfer could be discussed more extensively
- Discussion on / references to competing approaches based on feature distillation are not present


Remarks
------------

- This approach is similar in spirit to (https://arxiv.org/pdf/1910.10986.pdf, to be cited) where the authors try to align feature maps between past models and the current model. The general idea is still to "summarize" the previous model but only use feature maps instead of gradient attribution maps. I think one of the baseline missing in this paper is to use the feature maps themselves as targets for the distillation, would that be feasible ? How would that compare to distilling gradient attribution maps instead ? I feel the result can be stronger / more interesting if the authors include this baseline.

- It seems to me that distilling attributions of the previously trained models could limit backward-transfer, as the previous attributions acts like a sort of prior on the new information that can be learned. The prior seems "strong" as it doesn't change with the number of tasks: the model predicts the same attributions even if new information could indicate that the original "explanations" were not correct. Do you think backward transfer can be hurt by this ? Instead, could you maybe try to compute the XAI for *all* the examples in the buffer with the model estimated after task t, instead of computing only the XAI for current examples ? That would boil down to "filtering" the prior p(XAI_0) of attributions for task 0 with the posterior p(XAI_0 | 1, ..., t).

- I found quite hard to read Figure 2 and Figure 3. I'd suggest the authors to put in a Table the ablations results with respect to the different XAI methods and some of the most important differences in performance for Figure 3.

---

> ### Author Response · Authors · 2020-11-20
> **Responses to individual comments from AnonReviewer4**
>
> We would like to thank you for your time and feedback. Please see our responses below to the questions raised under "Cons" and "Remarks" in your comments. We will also upload the updated pdf shortly.
>
> 1. This approach, similar to all the regularization-based approaches in continual learning, is limited to the number of tasks it can scale to as they all have a fixed network architecture which does not change in size. Hence choosing a large memory budget for $M^{RRR}$ would potentially prevent the model from learning new tasks. We added this to the end of Section 3.2.
>
> 2. We have already cited Learning without memorizing(CVPR19) and learning without forgetting (ECCV16) and we added Encoder-based lifelong learning and  Adversarial Feature Alignment to our related work section. If the reviewer has specific other prior works in mind we hope they can let us know.
>
> 3. We would like to highlight the fact that in Grad-CAM, we are storing a linear combination of feature maps (**not their gradients**) up to the last convolutional layer in the architecture (Eq 1 in the paper). As requested, we ran an experiment where we performed the regularization on the feature maps generated by the last convolutional layer of ResNet18 in the few-shot CIL CUB200 experiment. Results are shown below (averaged over 3 runs) which show that RRR outperforms this type of regularization by a small margin. For easier comparison, we have included RRR and ER results alongside. We cited the suggested paper (https://arxiv.org/pdf/1910.10986.pdf) in which they have a combination of four objective functions one of which regularizes the output feature maps of the last convolutional layer whereas in RRR we regularize a linear combination of feature maps up to the last convolutional layer for a fixed set of samples stored from prior tasks.
>
> **ER**:
>
> [$67.8\pm0.8$ 	, $49.7\pm0.9$ 	, $41.7\pm0.8$ 	, $35.8\pm0.7$ 	, $31.4\pm0.9$ 	, $28.5\pm0.8$ 	, $25.5\pm0.8$ 	, $22.1\pm0.8$ 	, $21.8\pm0.6$, $22.5\pm1.1$ 	, $19.8\pm0.9$]
>
> **RRR:**
>
> [$67.8\pm0.8$ , $53.5\pm0.7$ , $45.6\pm0.6$ , $39.6\pm0.7$ , $35.3\pm0.9$ , $32.3\pm1.1$ , $29.4\pm0.9$ , $25.9\pm0.8$ , $25.7\pm0.6$ , $26.3\pm0.7$ , $23.6\pm0.7$]
>
> **Feature map of the last convolution:**
>
> [$67.8\pm0.8$ , $51.5\pm0.4$ , $44.3\pm0.4$ , $39.3\pm0.3$ , $33.6\pm0.8$ , $31.4\pm1.0$ , $27.7\pm0.7$ , $24.3\pm0.5$ , $24.5\pm0.4$ , $25.6\pm0.6$ , $23.3\pm0.6$]
>
> 4. Our objective function in Eq. 1 shows that we are computing the XAI on all the samples stored in $M^{RRR}$ and that is the main benefit of the RRR method that helps with increasing the backward-transfer.
>
> 5. We added tables to our supplementary material to show the numbers used to generate all of our plots.

---

### Decision · Program_Chairs · 2021-01-07
**Final Decision**

**Decision:**

Accept (Poster)

**Comment:**

The paper touches upon the problem of catastrophic forgetting in continual learning. The idea is to enhance experience reply by explanations of the decision/predictions made. Technically, this "Remembering for the Right Reasons" loss adds an explanation loss to continual learning. This is an interesting idea as also the reviewers agree on. I would like to encourage the authors to have consider a different abbreviation. RRR also stand for "Right for the Right Reasons" loss due to Ross et al.; the authors should use a different abbreviation and also mention the work of Ross et al. (Andrew Slavin Ross, Michael C. Hughes, Finale Doshi-Velez: Right for the Right Reasons: Training Differentiable Models by Constraining their Explanations. IJCAI 2017: 2662-2670). Moreover, it might actually be interesting in moving towards interactive learning here as well, because continual learning may also suffer from confounders. Moreover, there is also a connection to HINT (Ramprasaath Ramasamy Selvaraju, Stefan Lee, Yilin Shen, Hongxia Jin, Shalini Ghosh, Larry P. Heck, Dhruv Batra, Devi Parikh: Taking a HINT: Leveraging Explanations to Make Vision and Language Models More Grounded. ICCV 2019: 2591-2600) as it also aims at keeping explanations close to each other. Indeed, they use a ranking loss and do not consider continual learning. Overall, a simple method that is shown empirically to help improving existing replay methods for class-incremental learning.